# The impact of matches and travel on rugby players' sleep, wellness and training

**Michele Lo**[1], **Robert J. Aughey**[1], **William G. Hopkins**[1], **Nicholas Gill**[2,3], **Andrew M. Stewart**[1]*

**1** Institute for Health and Sport (iHeS), Victoria University, Melbourne, Australia, **2** Adams Centre for High Performance, University of Waikato, Tauranga, New Zealand, **3** New Zealand Rugby Union, Wellington, New Zealand

* Andrew.Stewart@vu.edu.au

**Data Availability Statement:** The values used to build the graphs can be retrieved here: https://osf.io/fupvx.

**Funding:** The authors received no specific funding for this work.

## Abstract

Matches and travel, which are common in professional team sports, may have a negative impact on players. The aim of this study was to quantify the impact on sleep, wellness and training of Super Rugby players. Sleep of 122 players from four teams was monitored using activity monitors for up to three nights before and after matches played at home and overseas. Wellness and internal training load (questionnaires) and external training load (GPS/accelerometer) were also recorded. Separate analyses were performed for each team using a general linear mixed model to estimate the mean effects of travel (translocation overseas and return to the home country) on sleep, wellness and training. The mean effects of matches on sleep and wellness on the nights before, of and after matches. were also estimated. Teams generally experienced small to large reductions in sleep and wellness when overseas; on return, sleep and wellness recovered somewhat. The impact of matches on sleep and wellness differed in magnitude and direction (large reductions to small increases) between teams. External load overseas and upon return was reduced for three of the four teams, whilst internal load was reduced for the three teams that measured it. The changes in sleep, wellness and training can be explained by a combination of travel- and match-related stressors that differed between teams. Teams should consider remediation strategies to mitigate the effects of travel.

## Introduction

Super Rugby was the most important Rugby Union competition outside Europe at a club level. The competition involved 15 teams from five different countries (Argentina, Australia, Japan, New Zealand, and South Africa) playing each other home or away during the season. As such, teams had to travel frequently throughout the year. Even if international travel is common for several sports, there was no other competition with such a large travel demand for athletes.

Frequent air travel can have detrimental effects on athletes' sleep [1], training [2], and wellness [3]. The negative effects of travel are mostly related to travel fatigue and jet lag [4]. Travel fatigue is a state of persistent weariness, recurrent illness and changes in mood that can occur

**Competing interests:** The authors have declared that no competing interests exist.

after a single trip and accumulate over time [4]. Jet lag is the desynchronization between the body clock and the external clock that occurs after traveling across time zones, with a shift of the external clock and a change on the cues that drive the body clock [5, 6]. Jet lag symptoms include sleep disturbances, fatigue, and depressed mood [5].

Sleep is a behavioural state in which an individual is perceptually disengaged from, and unresponsive to the environment [7]. Although sleep functions are not fully understood, it is generally accepted that sleep can positively influence mood, wellbeing, and permits recovery from previous wake phases [7]. Athletes require more sleep than other populations and athletes believe good sleep is important to perform at their best [8]. However, an athlete's sleep is usually disrupted [7] and of poor quality [8]. Rugby players are also exposed to additional stressors including frequent trans-meridian travel that are likely to negatively impact their sleep and quality of life [9].

The purpose of this study was to determine the influence of trans-meridian travel and matches on the sleep, wellness, and training of players from four Super Rugby teams during the 2017 Super Rugby season.

## Material and methods

Four professional Super Rugby teams, two from New Zealand and two from South Africa, were approached, and a total of 122 male players (age range 18–35 y) gave written informed consent to participate. The study, a prospective observational study of changes in sleep, wellness and training with travel and occurrence of matches in four rugby teams, was conducted during the 2017 Super Rugby season. All players' data were de-identified prior to analysis. The study was conducted according to the Declaration of Helsinki and approved by the authors' institutional Human Research Ethics Committee. Data were collected in multiple periods and at different times during the season. Before the beginning of each monitored period, players spent at least one week in their own country. Thus, participants would have already recovered from any travel that might have occurred outside of the monitored periods. Details on the monitoring agenda are presented in Table 1. Participants were monitored from three days prior to and up to three days post matches played in their own country. For matches played overseas, players were monitored from at least one day prior to departure. When possible, teams were also monitored for at least one day upon return. Each team was monitored for at least three matches (two in their own country and one overseas). Differences in the monitoring agenda are due to logistic reasons. A total of 19 matches, including 8 overseas, was monitored. For the matches played overseas and upon return, the number of time zones crossed and the number of days following travel were included in the analysis. To account for the difference in day length during trans-meridian flights, the travel day was added to the first day upon arrival and then scaled to a normal 24 hours. The number of time zones crossed was calculated as the difference between the time zones of the city where a match was played and of the previous location.

Participants' sleep/wake behaviour was monitored using sleep diaries, where participants recorded the start and end time for all sleep periods, and activity monitors (Philips Respironics, Actiwatch 2, Murrysville, Pensylvania). The activity monitors are wrist-watch accelerometer devices that recorded participants' movement. Participants wore the device on the same wrist for the duration of the study, excluding training and matches. Sleep was assessed in 1-min epochs: participants were considered asleep when the diary indicated they were in bed and the activity recorded by the watches was sufficiently low (less than 40 activity events per 1-min epoch) to indicate absence of movement [10].

**Table 1. Temporal order of matches around which sleep was monitored for each team, showing location where each match was played, number of time zones crossed and direction of travel to reach the match venue.**

| Team | Match | Match location | Number of time zones crossed | Travel direction |
|------|-------|----------------|------------------------------|------------------|
| A | 1 | Overseas | 2 | West |
|   | 2 | Away | 2 | East |
|   | 3 | Home | 0 | - |
|   | 4 | Overseas | 10 | West |
|   | 5 | Home | 10 | East |
| B | 1 | Overseas | 3 | West |
|   | 2 | Home | 3 | East |
|   | 3 | Away | 0 | - |
|   | 4 | Home | 0 | - |
|   | 5 | Overseas | 10 | West |
|   | 6 | Overseas | 6 | East |
|   | 7 | Home | 4 | East |
| C | 1 | Away | 0 | - |
|   | 2 | Home | 0 | - |
|   | 3 | Overseas | 5 | West |
| D | 1 | Home | 0 | - |
|   | 2 | Overseas | 10 | East |
|   | 3 | Overseas | 0* | -* |
|   | 4 | Away | 0 | - |

*Match played in the same time zone as for the previous match.

Dashed lines indicate non-monitored periods.

Wellness and training load were recorded by each team as part of their normal monitoring routine. Each team used a Likert scale for wellness and, although some of the items where in common, others were exclusive to a specific scale. As such items on each scale differed in number and wording; to account for these differences, all responses were rescaled linearly to a score ranging from a minimum possible of 0 to a maximum possible of 100. External load was monitored by each team using different GPS/accelerometer devices. Two teams used VX340b and VX350 units sampling at 10 Hz (VX Sport, Lower Hutt, New Zealand), one team used Catapult OptimEye S5 units sampling at 10 Hz (Catapult Innovations, Melbourne, Australia) and one team used SPI HPU units sampling at 10 Hz (GPSports, Canberra, Australia). Although the validity and reliability of these devices were not directly assessed for logistic reasons, all devices, excluding the VX350, have shown acceptable accuracy for team sport use or in a laboratory setting [11–13]. Team C provided external load data only as a weekly average of all training sessions and therefore their data were not included. Teams used different customized speed bands to evaluate external load. Total distance (metres run in total) and relative distance (m.min$^{-1}$) were retrieved and analysed from all three teams as these measures are not based on speed bands. Two teams (Team A and D) used the same threshold to define high-speed distance (metres run above 4.17 m.s$^{-1}$) and therefore their data were included. Team B did not provide any high-speed distance data.

Internal load was calculated by each team as a product of the session rating of perceived exertion (sRPE) and the duration of each session in minutes [14]. Two teams directly provided internal load data, while one team provided both sRPE score and training duration that permitted the calculation of internal load. Team A provided internal load as an average from a different sample of players for each training session and therefore their data were not included.

## Statistical analysis

Data were imported into the Statistical Analysis System (version 9.4, SAS Institute, Cary, NC). The analyses were performed with a general linear mixed model (Proc Mixed) to estimate changes in sleep, wellness, internal load, and external load whilst overseas and upon return. Separate analyses were performed for each team, because differences between teams in the number of trips and in travel and recovery periods prevented development of a single model that would do justice to each team. For all outcomes, fixed effects predicted a mean value for all normal days at home, for each day following travel overseas and for each day following return from overseas. Dummy variables were included to estimate the effects of peri-match days (the day before, day of and day after a match) for all matches played only in their own country. These effects were subtracted from the estimates for peri-match days following travel to and from overseas to visualise the pure effects of travel for all travel days. The mean effects of travel were calculated for all the monitored days following travel to and from overseas, excluding peri-match days. For the sleep analysis, a combined effect of all peri-match nights (sleep balance) was also included.

A within-player random effect, given by a dummy variable with a value of 0 for normal nights and 1 for any night including and following travel, was included to estimate individual differences in the response to travel. This effect and a player random effect were assigned an unstructured covariance matrix to allow for a correlation between the two effects. This correlation was evaluated as the effect of 2 SD [15] of the true differences between players (the square root of the variances given by the player random effect) on the individual differences; from first principles this effect is given by twice the covariance divided by the true between-player standard deviation. With sufficient data, confidence limits for this effect could be derived by bootstrapping.

A residual variance was specified to allow for individual changes within each player between normal days in their own country. Different residual variances were also specified for each travel day to allow for individual changes between travel days (not accounted for by the within-player random effect). For sleep and training variables, the between-player variance and the residual variance for normal days were combined to estimate a between-player observed standard deviation for normal days, which was then averaged across the four teams and rounded to the closest meaningful integer (75 minutes for sleep) to assess the magnitude of the effects via standardization. The magnitude thresholds for small, moderate, large and very large effects were 0.2, 0.6, 1.2 and 2.0 of the observed standard deviations of each variable [15]. For wellness, standardisation would have provided an unrealistic smallest important effect of 2 units on the 0–100 scale. A smallest important of 10 units, based on 10% of a psychometric scale [16], is arguably unrealistically large. A value of 5 units was therefore chosen as a compromise, with the other magnitude thresholds provided by the factors of the standardised scale [15]. The magnitudes of the standard deviations representing the individual responses to travel were evaluated by halving the thresholds [17].

The potential mediating effect of sleep on wellness would normally be investigated by including sleep in the model to determine its effect after adjustment for the other predictors and to determine any reductions in the effects of the other predictors after adjustment for sleep (i.e., holding sleep constant) [15]. There were insufficient data for such an analysis here, so a simpler mediation analysis was performed, in which sleep variables were the only predictors in the model. The fixed effects were the mean sleep duration of each player for all normal nights in their own country (to estimate the between-player effect of sleep) and sleep duration rescaled to a mean of zero for each player (to estimate the within-player effect of sleep). This rescaled variable was included as a random effect, along with player identity, in an unstructured

covariance matrix to allow for individual differences in mean wellness and in the mediating effect of sleep. Residual variances were specified as in the previous model. The between-player and within-player effects of sleep were evaluated for twice the relevant SDs: the between-player SD of the players' mean sleep duration on normal nights, and the mean of the within-player SD for each team. Separate analyses were performed for each team.

Uncertainty in the outcomes was expressed as 90% compatibility limits. Decisions about magnitudes accounting for the uncertainty were based on one-sided interval hypothesis tests, according to which a hypothesis of a given magnitude (substantial, non-substantial) is rejected if the 90% compatibility interval falls outside that magnitude [18, 19]. P values for the tests were therefore the areas of the sampling distribution of the effect (t for means, z for variances) falling in the hypothesized magnitude, when the distribution is centered on the observed effect. Hypotheses of inferiority (substantial negative) and superiority (substantial positive) were rejected if their respective p values ($p_-$ and $p_+$) were <0.05; rejection of both hypotheses represents a decisively trivial effect in equivalence testing. When only one hypothesis was rejected, the p value for the other hypothesis, when >0.25, was interpreted as the posterior probability of a substantial true magnitude of the effect in a reference-Bayesian analysis with a minimally informative prior [20] using the following scale: >0.25, possibly; >0.75, likely; >0.95, very likely; >0.995, most likely [15]; the probability of a trivial true magnitude ($1-p_--p_+$) was also interpreted, when >0.25, with the same scale. A non-informative prior was justified in the present study, because there was negligible shrinkage of effects when a normally distributed weakly informative prior was applied in a semi-Bayesian analysis [20] promoted by Greenland [21]. The greatest shrinkage occurred with the largest effect on sleep (-138, ±46 min): with a normally distributed weakly informative prior centered on 0 and with a 90% compatibility interval excluding extremely large effects (±300 min), the posterior (-135, ±46 min) represents negligible shrinkage. Additionally. a non-informative prior does not bias effects [20]. Probabilities were not interpreted for effects with inadequate precision at the 90% level, defined by failure to reject both hypotheses ($p_->0.05$ and $p_+>0.05$). To account for inflation of error, effects with adequate precision at the 99% level ($p_-<0.005$ or $p_+<0.005$) were highlighted. The hypothesis of non-inferiority (non-substantial-negative) or non-superiority (non-substantial-positive) was rejected if its p value ($p_{N-} = 1-p_-$ or $p_{N+} = 1-p_+$) was <0.05, representing a decisively substantial effect in minimal-effects testing: very likely or most likely substantial.

## Results

Changes from the mean normal sleep time for each team on each monitored night spent overseas and upon return are showed in Fig 1. The average sleep on normal nights and the effects of travel on sleep are presented in Table 2; as shown, three of the teams, on average, had a small to moderate decrease in their sleep time (up to one hour) when overseas, while one team (Team A) experienced a small increase. On return from travel two teams showed trivial changes in their sleep, while the team that was monitored for only one night upon return (Team C) experienced a moderate sleep loss. The individual responses showed a small to moderate positive response to travel, whilst the effects of 2 between-player SD on such responses were negative and trivial to moderate.

All the effects were adjusted for any peri-match effect, which are presented in Table 3. As shown, on the night before a match, three teams had a small increase in their sleep time while one team had a small decrease (Team D). The night following a match three teams had a moderate to large decrease on their sleep time whilst one team (Team A) had a small increase. On the following night, three teams had a trivial to moderate decrease in their sleep, while one team had a small increase (Team C). Sleep balance, represented by the amount of sleep gained

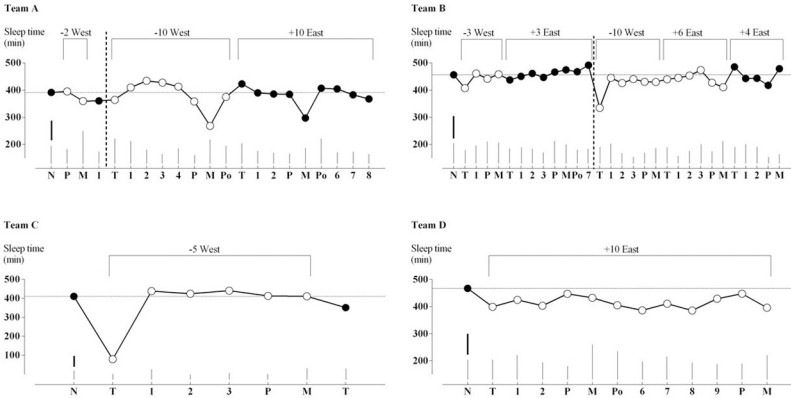

**Fig 1. Estimated sleep time after adjusting for peri-match effects during nights spent overseas and upon return compared to normal nights.** Thin lines are within players SD for each night. Thick lines are observed between players SD for each team. Dotted horizontal lines are normal sleep duration for each team. Dashed vertical lines represent a time gap during data collection. Square brackets report number of time zones crossed and direction of travel. Black symbols are home nights, empty symbols are overseas nights. X-axis abbreviations: N, normal nights; P, pre-match nights; M, match nights; Po, post-match nights; T, travel nights; numbers represent nights following travel.

and lost over the peri-match nights, was negative for three teams (small to large loss) and positive for one team (Team A, moderate gain).

Changes from the mean normal wellness for each team on each monitored day spent overseas and upon return are shown in Fig 2. The average wellness on normal days and the effects of travel on wellness are presented in Table 4. As shown, when overseas, all teams experienced

**Table 2. Sleep on normal nights, changes in sleep for travel nights (overseas and upon return), individual responses to travel (expressed as an SD estimated across all travel nights), and effects of 2 between-player SD on individual responses.** All units are minutes.

| Team | Sleep mean ± SD | Travel changes, ±90%CL | | | Effects of 2 SD[a] |
|---|---|---|---|---|---|
| | | Overseas | Return | Individual responses | |
| A | 392 ± 72 | **18, ±17 S**[*][0] | 1, ±14 T[00] | **26, ±14 M**[***] | -26 S |
| B | 456 ± 70 | **-37, ±16 S**[***] | 0, ±17 T | 24, ±26 M[**] | -39 S |
| C | 411 ± 57 | **-65, ±12 M**[****] | **-59, ±24 M**[****] | 17, ±19 S[**] | -10 T |
| D | 467 ± 66 | **-62, ±19 M**[****] | n/a | **39, ±20 M**[****] | -52 M |

CL, compatibility limits.

[a]The SD were 30, 27, 33, 17 for Team A, B, C, D respectively (true between-player SD from the mixed model). CL for these effects were not available.

Observed magnitude: T, trivial; S, small; M, moderate; L, large.

Reference-Bayesian likelihoods of true substantial change:

[*]possibly;

[**]likely;

[***]very likely,

[****]most likely;

[***] and [****] indicate rejection (p <0.05 and <0.005 respectively) of the non-superiority or non-inferiority hypothesis.

Reference-Bayesian likelihoods of true trivial change:

[0]possibly;

[00]likely;

[000]very likely,

[0000]most likely.

Effects in **bold** have adequate precision at the 99% level (rejection of the superiority or inferiority hypothesis, p<0.005).

Effects with CL but without likelihoods have inadequate precision.

n/a indicates data were not available.

**Table 3. Changes in sleep for peri-match nights (including sleep balance) for each monitored team ±90% compatibility limits.** All units are minutes.

| Team | Pre-match | Match | Post-match | Balance |
|---|---|---|---|---|
| A | **41, ±20 S**\*\*\* | 17, ±21 S\*[0] | -12, ±20 T[0]\* | **46, ±39 M**\*\* |
| B | 19, ±23 S\*[0] | **-80, ±23 M**\*\*\*\* | -19, ±29 S\*[0] | **-81, ±51 M**\*\*\* |
| C | **19, ±14 S**\*[0] | **-60, ±14 M**\*\*\*\* | **19, ±15 S**\*[0] | -22, ±30 S\*[0] |
| D | -26, ±19 S\*\* | **-61, ±20 M**\*\*\*\* | -51, ±26 M\*\*\* | **-138, ±47 L**\*\*\*\* |

Observed magnitude: T, trivial; S, small; M, moderate; L, large.

Reference-Bayesian likelihoods of true substantial change:

\*possibly;

\*\*likely;

\*\*\*very likely,

\*\*\*\*most likely;

\*\*\* and \*\*\*\* indicate rejection (p <0.05 and <0.005 respectively) of the non-superiority or non-inferiority hypothesis.

Reference-Bayesian likelihoods of true trivial change:

[0]possibly;

[00]likely;

[000]very likely,

[0000]most likely.

Effects in **bold** have adequate precision at the 99% level (rejection of the superiority or inferiority hypothesis, p<0.005).

a moderate to large decrease in their perceived wellness. On return from travel, the observed changes in wellness were smaller. The individual responses to travel were small and clearly substantial for two teams but trivial and only possibly small for one team. One team had a negative SD consistent with no individual responses, given its uncertainty. The effects of 2 between-player SD on individual responses were negative and trivial to small in magnitude. All the effects were adjusted for any peri-match effect, which are presented in Table 5. As

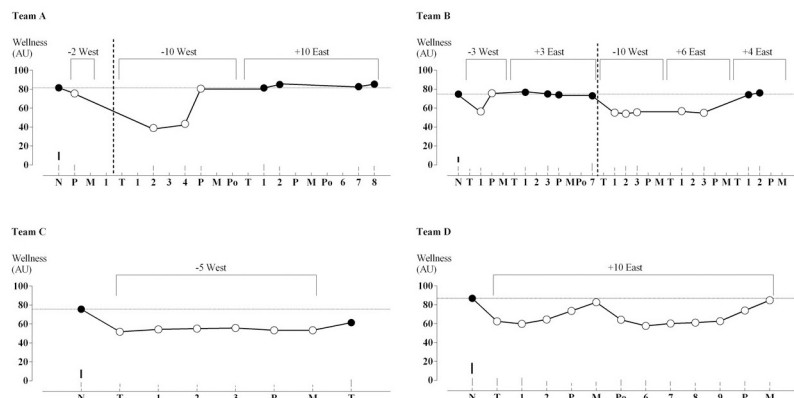

**Fig 2. Estimated wellness after adjusting for peri-match effects during days spent overseas and upon return compared to normal days.** Thin lines are within players SD for each day. Thick lines are observed between players SD for each team. Dotted horizontal lines are normal wellness scores for each team. Dashed vertical lines represent a time gap occurred during data collection. Square brackets report number of time zones crossed and direction of travel. Black symbols are home days, empty symbols are overseas days. X-axis abbreviations: N, normal nights; P, pre-match nights; M, match nights; Po, post-match nights; T, travel nights; numbers represent days following travel.

**Table 4. Wellness on normal nights (range 0 to 100), changes in wellness for travel days (overseas and upon return), individual responses to travel (expressed as an SD estimated across all travel days), and effects of 2 between-player SD on individual responses.** All units are arbitrary units.

| Team | Wellness mean ± SD | Travel changes, ±90%CL | | | Effects of 2 SD[a] |
|---|---|---|---|---|---|
| | | Overseas | Return | Individual responses | |
| A | 81.6 ± 8.8 | **-40.3, ±1.5 L****** | **2.1, ±1.2 T0000** | **-1.7, ±1.8 T**** | -0.7 T |
| B | 47.9 ± 6.2 | **-19.5, ±1.5 M****** | **-5.3, ±1.5 S*0** | **2.5, ±2.3 T0*** | -0.5 T |
| C | 75.7 ± 9.7 | **-21.4, ±2.0 M****** | **-14.2, ±3.3 S****** | **6.1, ±1.8 S****** | -11.2 S |
| D | 86.9 ± 11.8 | **-25.7, ±2.1 M****** | n/a | **5.0, ±1.7 S***** | -1.5 T |

CL, compatibility limits.

[a]The SD were 6.1, 3.9, 8.2, 8.3 for Team A, B, C, D respectively (true between-player SD from the mixed model). CL for these effects were not available.

Observed magnitude: T, trivial; S, small; M, moderate; L, large.

Reference-Bayesian likelihoods of true substantial change:

*possibly;

**likely;

***very likely,

****most likely;

*** and **** indicate rejection (p <0.05 and <0.005 respectively) of the non-superiority or non-inferiority hypothesis.

Reference-Bayesian likelihoods of true trivial change:

0possibly;

00likely;

000very likely,

0000most likely;

000 and 0000 indicate rejection (p <0.05 and <0.005 respectively) of the superiority and inferiority hypotheses.

Effects in **bold** have adequate precision at the 99% level (rejection of the superiority or inferiority hypothesis, p<0.005).

n/a indicates data were not available.

**Table 5. Changes in wellness (range -100 to 100) for peri-match days for each monitored team ±90% compatibility limits.** All units are arbitrary units.

| Team | Pre-match | Match | Post-match |
|---|---|---|---|
| A | **-30.8, ±2.1 L****** | n/a | n/a |
| B | **-19.8, ±2.0 M****** | n/a | n/a |
| C | **2.4, ±1.2 T0000** | **3.1, ±1.3 T000** | **-1.8, ±1.3 T0000** |
| D | **-11.5, ±2.2 S****** | **-20.9, ±2.1 M****** | **-0.5, ±3.0 T000** |

Observed magnitude: T, trivial; S, small; M, moderate; L, large.

Reference-Bayesian likelihoods of true substantial change:

*possibly;

**likely;

***very likely,

****most likely;

*** and **** indicate rejection (p <0.05 and <0.005 respectively) of the non-superiority or non-inferiority hypothesis.

Reference-Bayesian likelihoods of true trivial change:

0possibly;

00likely;

000very likely,

0000most likely;

000 and 0000 indicate rejection (p <0.05 and <0.005 respectively) of the superiority and inferiority hypotheses.

Effects in **bold** have adequate precision at the 99% level (rejection of the superiority or inferiority hypothesis, p<0.005).

n/a indicates data were not available.

shown, the night before a match, three teams had a small to large decrease in their wellness while one team had a trivial increase (Team C). The night following a match Team C experienced a trivial increase in wellness whilst Team D experienced a moderate decrease. On the following night both Team C and D experienced a trivial decline in their perceived wellness.

The between-player mediating effects of sleep on wellness were not consistent across the teams, ranging from likely substantial negative to likely substantial positive. The within-player mediating effects were more precisely defined: very likely to most likely trivial for three teams and a small, most likely positive effect for one team (Team C). Data are presented in S1 Table.

Training overseas for Team A showed a large to extremely large (clearly substantial) increase in total distance and total distance at high-speed and a likely small reduction in relative distance; inspection of their data showed a single, very intense training session on one of their overseas trips and modest reductions otherwise. Team D showed a large clearly substantial reduction in relative distance. All the other teams showed trivial to small reductions in all external load variables whilst overseas. Upon return, Team A showed a likely trivial increase in total distance above baseline, while the changes for the other measures and the changes for all measures for all the other teams were reductions ranging from likely trivial to most likely moderate. When overseas, the three teams recording internal load showed small to moderate reductions ranging from possibly to most likely substantial; for the two teams providing data upon return, there were possibly or likely trivial reductions relative to baseline. Data are presented in S2 Table.

## Discussion

Travel was associated with substantial sleep deprivation for three of the teams when overseas, which can be explained by travel fatigue, jet lag and a disruption of the normal sleep habit (sleeping in a non-familiar environment and sharing room with a team-mate). Travel fatigue may have had a greater impact on sleep during the flight and the first night upon arrival. After the first night, sleep disruption is due mostly to jet lag, as a full night of rest is usually enough to recover from the effects of travel fatigue [1, 4]. Players from Team A slept more than normal whilst overseas. This team also had the lowest mean sleep at home. According to information reported by the coaching staff, this team scheduled home training early in the morning, and players had to wake early to commute to the training facility. Apparently, travel away from home provided players in this team with an opportunity to catch up on sleep loss.

Westward travel is considered more detrimental than eastward travel as the human body shows a natural tendency to drift slightly each day and it is easier for the body to cope with a delay, which occur travelling westward, than an advance in time, which occur travelling eastward [22]. However, after travelling either east or west, players tended to sleep longer for each night following travel. Humans need approximately one day per time zone crossed to recover from jet lag [6], but the rate and velocity of recovery can be accelerated by implementing specific strategies, such as melatonin supplementation [5, 22]. Super Rugby teams reportedly used travel strategies that may have reduced sleep loss when overseas [23]. For two of the three teams monitored upon return from overseas, sleep returned to normal values. A possible explanation is that players returned to a familiar environment. The other team was monitored only for one day after travel, and fatigue may have affected their first night of sleep. The individual players' responses to travel were substantial in two teams, and there was some evidence of substantial responses in the other two, which could reflect sensitivity to travel fatigue and jet lag or differences between individual approaches to travel issues. The substantial observed negative association of individual responses with normal sleep for three of the teams suggests that players who sleep more than average should take more care about avoiding sleep loss after long-haul travel, especially as they could be more 'disturbed' by sleep deprivation [24]. Sleep

hygiene, nutritional interventions and napping are all strategies that can be effective in avoiding sleep loss in general and after trans-meridian travel [7, 8].

The analyses of the changes in sleep when overseas involved quantification and adjustment for the changes in sleep on pre-match, match, and post-match nights. The night before matches, there was some evidence of players sleeping more than normal, suggesting an attempt to increase sleep time (for example, going to bed earlier than usual) to potentially improve preparedness for the match [25, 26]. Only Team D showed an opposite trend. Inspection of their match schedule showed that this team was monitored before matches played against direct competitors for a spot in the play-offs, so pre-match anxiety may have affected their sleep. The negative effect of playing a match on the subsequent night of sleep may be explained by the fact that Super Rugby matches are usually scheduled for late evening, and the post-match routine for the players (shower, dinner, media window, etc.) delay their time to bed [27]. Furthermore, the stress accumulated during the match, soreness, and consumption of caffeine or other 'energetic' drinks have a negative impact on the quality and quantity of sleep after a match [7]. Only Team A players increased their sleep somewhat after a match. According to information reported by the coaching staff, this team did not train the day after a match. The chance to sleep more than usual may have provided players in this team with another opportunity to catch up on sleep loss. All but one team experienced a reduction in their sleep during post-match nights, which is possibly an aftermath of all the match-related issues. The final sleep balance for all the peri-match nights was substantially negative in two teams, and there was some evidence of a substantial sleep loss for another team. Given the hectic schedule of Super Rugby, with matches played every week and a short recovery window in between, players may constantly lose sleep time during the season. This chronic sleep deprivation has the potential to affect health and wellbeing of the players [28]. Only players from Team A had a somewhat positive sleep balance. However, their particularly low baseline sleep suggests they may suffer of chronic sleep deprivation to a larger extent than players from other teams.

Travel across time zones had a substantial negative effect on players' wellness. Although eastward travel is generally seen as more detrimental than westward travel [22], wellness decreased for all teams when travelling overseas, regardless of the direction, and tended to improve upon return. A possible explanation is that well-being is complex and multidimensional [29], and the combined effect of 'home-sickness', match and other factors [30] had a larger impact on wellness than direction of travel. The individual players' responses to travel were substantial for two teams, which provides some evidence of sensitivity to travel or differences between individual approaches to travel, albeit to a lesser extent than the individual responses to sleep. The changes in wellness when overseas involved quantification and adjustment for changes in wellness on peri-match days. Pre-match anxiety evidently had a role in the reduction in wellness for three of the teams on the days leading to a match and for one of the teams on the day of the match. A rebound from pre-match anxiety could then explain the return of wellness to only a trivial difference from baseline on the post-match day for the two teams who provided post-match data. Athletes who feel well may perform better [31] and may be less prone to injury [2], so team management should investigate ways to improve well-being when the team is overseas.

The uncertainties in the mediating effects of sleep on wellness between players are consistent with no useful relationship between the habitual levels of sleep and wellness in any of the teams. Investigation of such between-player relationships requires numbers of players much larger than those in any squad to provide meaningful information. On the other hand, within players, one team (Team C) showed a substantial small relationship in the expected direction of more sleep leading to better wellness [30]. There was strong evidence for a lack of relationship in the other teams, which may be related to poorly reported sleep and/or bias in the

athletes reporting of their wellness or an inherent lack of validity in the different wellness scales used by each team. Future investigation of this relationship should carefully monitor and try to increase the athletes' adherence to the research protocols.

Travel overseas had a somewhat negative effect on training load. Only one team showed a substantial increase in external load, due to a single, hard training session. Internal load was also somewhat lower than baseline when overseas, showing that players perceived their training sessions as less intense. Teams tend to modify training when overseas, for instance planning 'technical/tactical' rather than physical sessions (e.g., less tackles or contact phases than usual), to reduce the stress of travel on players [2]. One of the teams (Team A) evidently used travel overseas to increase training and even on their return, but they reduced their high-speed training on return. For the other teams, there were insufficient data to determine the extent to which they made up for any training lost overseas in the days following their return.

There are several limitations in this study related to the quantity and quality of data. This investigation was essentially four case studies of Super Rugby teams, so conclusions about effects of travel on Super Rugby teams generally, and about individual differences between teams, would require study of at least all teams involved in the competition. Each team used a different wellness scale. The scales were not validated and some of the scales included different components, compared to the others. The limitations of the tools used to measure wellness could have had an impact on our results. Future study should monitor wellness with standardised and validated tools to accurately address the changes in wellness following travel. The analyses also had to be simplified in several ways because of the limited data provided by each team. First, it was not possible to discriminate between travel and return when analysing individual responses to travel. Secondly, travel effects were estimated by accounting for peri-match effects for matches in each team's own country, but peri-match effects could be substantially different when the players are overseas. Finally, several sleep variables, including sleep onset, were not included in an attempt to reduce the inflation of error that can arise from analysing multiple effects and to account for some missing data in the sleep dairies. Future studies should monitor more teams for longer periods, including more overseas matches. Other sleep variables should be included where possible to determine the extent to which changes in sleep duration are related to changes in sleep onset and other sleep disturbances.

The findings of this research suggest that players in four Super Rugby players suffer reduced wellness and an overall sleep deficit when they travel overseas. As trans-meridian travel appears to affect players' sleep, teams should implement strategies such as melatonin supplementation and light exposure to reduce the effect of jet lag [5]. A correct sleep hygiene could also help players in catching up with the sleep loss they may experience throughout the season and following travel [7]. As there was some evidence of substantial individual responses, teams should carefully monitor the sleep of their players with particular attention to those who sleep more than average, as they may suffer more sleep disruption. Future investigations should aim to identify the best strategies to increase athletes' sleep and wellness when travelling or returning from overseas.

## Supporting information

**S1 Table. Mediating effects of sleep on wellness.**
(DOCX)

**S2 Table. Changes in training load overseas and upon return.**
(DOCX)

## Author Contributions

**Conceptualization:** Michele Lo, Robert J. Aughey, Nicholas Gill, Andrew M. Stewart.

**Formal analysis:** Michele Lo, William G. Hopkins.

**Investigation:** Michele Lo.

**Methodology:** Michele Lo, William G. Hopkins.

**Project administration:** Michele Lo, Andrew M. Stewart.

**Supervision:** Robert J. Aughey, Andrew M. Stewart.

**Writing – original draft:** Michele Lo.

**Writing – review & editing:** Robert J. Aughey, William G. Hopkins, Nicholas Gill, Andrew M. Stewart.

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
