## [Decision Letter · Decision Letter 0]

7 Jun 2021

PONE-D-21-08417

The impact of matches and travel on rugby players’ sleep, wellness and training.

PLOS ONE

Dear Dr. Stewart,

Thank you for submitting your manuscript to PLOS ONE. After careful consideration, we feel that it has merit but does not fully meet PLOS ONE’s publication criteria as it currently stands. Therefore, we invite you to submit a revised version of the manuscript that addresses the points raised during the review process.

Please, address the expert reviewer concerns but consider also that I will need a second reviewer to complete the peer review process.

We look forward to receiving your revised manuscript.

Kind regards,

Daniel Boullosa

Academic Editor

PLOS ONE

Journal Requirements:

2. Please include captions for your Supporting Information files at the end of your manuscript, and update any in-text citations to match accordingly. Please see our Supporting Information guidelines for more information: http://journals.plos.org/plosone/s/supporting-information

Reviewers' comments:

Reviewer's Responses to Questions

**Comments to the Author**

1. Is the manuscript technically sound, and do the data support the conclusions?

Reviewer #1: Partly

2. Has the statistical analysis been performed appropriately and rigorously? 

Reviewer #1: I Don't Know

3. Have the authors made all data underlying the findings in their manuscript fully available?

Reviewer #1: Yes

4. Is the manuscript presented in an intelligible fashion and written in standard English?

Reviewer #1: Yes

5. Review Comments to the Author

Reviewer #1: General Comments

Thank-you for the opportunity to review this manuscript. The work is very well written and the attempt to quantify responses in real world high performance environments, despite the limitations, is a real positive.

I've made some specific comments below that the author's may wish to consider.

Specific Comments

p5, Table 1: the number of time zones crossed is informative but it would also help to provide the reader with context if this table contained data on flight durations and distance.

p5, L86: as the authors are aware, the software associated with the activity monitors used in this study can determine a range of variables in addition to sleep duration (e.g. sleep onset latency, sleep efficiency % etc.). Whilst duration should clearly be included, I feel like much more insight could be gained by the analysis and inclusion of extra variables. Furthermore, the reporting of variables from the sleep diary such as bed time and wake up time could be obvious additions. This is probably the major limitation of the work but if the data exists, it would substantially enhance the manuscript.

p6 L96: The specifics of the individual team wellness scales would benefit from further explanation. Given the "items differed in number and wording", I'm not completely clear on whether they are essentially considering the same wellness components (e.g. soreness, fatigue etc. etc.) or something different (which obviously impacts the interpretation of the impact of travel/sleep on wellness). Whilst many researchers (me included) have published work involving customised wellness questionnaires, the lack of validation of these inventories should be acknowledged. They may well prove to have validity similar to established tools such as the DALDA, MTDS, ARSS/SRSS, but until then we should probably be cautious with any findings.

p6, L103: it might be worth checking the specifications of the SPI HPU units. As far as I know, they use a GPS chip that samples at 10Hz and this data is supplemented with information from the accelerometers. Their suggestion that the device samples at 15Hz might not be technically correct.

p8, L154: see previous comments regarding wellness.

p11, Table 2 (and all subsequent tables): whilst I completely understand the challenge in presenting statistical analysis results, the tables are potentially a little "busy". Trying to capture multiple levels of analysis with various symbols and formatting makes for somewhat confusing reading in the first instance. For example, "0" and "*" are each used for specific purposes. However, I acknowledge that after a few reads, the tables makes sense.

On a more general statistical analysis front, whilst I see the merit in levels of likelihoods of a true substantial change, this type of approach has been heavily criticised. My understanding is that the suggestion has been to either conduct a fully Bayesian analysis or a one-sided hypothesis test as an alternative(as has been done here). I'm aware that the authors are confident in suggesting that the method used is "Reference-Bayesian" but it should be acknowledged that not everyone seems to support this and I'm not aware of this journal's policy on this issue.

p16, L315: it would be beneficial for the authors to provide some suggestions as to how the players who sleep more than average should take more care about avoiding sleep loss. What strategies could they employ?

p17: as mentioned earlier, the discussion relating to the wellness results would benefit from being considered within the context of the potential limitations of the measurement tool.

p18, L358: the sentence beginning "Athletes who feel well............" is a little to certain in my view. Consider rewriting to acknowledge the uncertainty about these links.

p18, L367: is it possible that an inherent lack of validity in the wellness scale is at least partly responsible for this lack of relationship?

p19, L390: this sentence relates to my previous point about additional sleep variables but I'm not completely sure what's meant here. Numerous variables can be obtained from the activity monitors

p19: the inclusion of a limitations section is a strength and I believe that even with these limitations acknowledged that real world data on multiple teams is worth publishing. A general comment relating to the discussion section is that it parts it feels a tiny bit superficial. I know there is the limitation of word count, but there might be merit in more detailed discussion about the implications of the findings (I accept this can be limited due to the largely observational nature of the study) and perhaps some more information on possible strategies to mitigate the negative effects of this type of travel schedule and resultant changes to sleep.

6. PLOS authors have the option to publish the peer review history of their article (what does this mean?). If published, this will include your full peer review and any attached files.

Reviewer #1: No

---

## [Author Response · Author response to Decision Letter 0]

30 Sep 2021

Response to reviewers document attached to submission

---

## [Decision Letter · Decision Letter 1]

28 Oct 2021

PONE-D-21-08417R1The impact of matches and travel on rugby players’ sleep, wellness and training.PLOS ONE

Dear Dr. Stewart,

Thank you for submitting your manuscript to PLOS ONE. After careful consideration, we feel that it has merit but does not fully meet PLOS ONE’s publication criteria as it currently stands. Therefore, we invite you to submit a revised version of the manuscript that addresses the points raised during the review process.

We look forward to receiving your revised manuscript.

Kind regards,

Daniel Boullosa

Academic Editor

PLOS ONE

Journal Requirements:

Reviewers' comments:

Reviewer's Responses to Questions

**Comments to the Author**

1. If the authors have adequately addressed your comments raised in a previous round of review and you feel that this manuscript is now acceptable for publication, you may indicate that here to bypass the “Comments to the Author” section, enter your conflict of interest statement in the “Confidential to Editor” section, and submit your "Accept" recommendation.

Reviewer #1: All comments have been addressed

Reviewer #2: (No Response)

2. Is the manuscript technically sound, and do the data support the conclusions?

Reviewer #1: Partly

Reviewer #2: (No Response)

3. Has the statistical analysis been performed appropriately and rigorously? 

Reviewer #1: I Don't Know

Reviewer #2: Yes

4. Have the authors made all data underlying the findings in their manuscript fully available?

Reviewer #1: Yes

Reviewer #2: No

5. Is the manuscript presented in an intelligible fashion and written in standard English?

Reviewer #1: Yes

Reviewer #2: Yes

6. Review Comments to the Author

Reviewer #1: Thank-you for addressing my comments on the original manuscript.

The responses and modifications seem reasonable to me. Reviewers and authors may sometimes have a different perspective, but in my view if the authors have provided justification for the approach then I'm happy.

Reviewer #2: To the author,

the manuscript is generally very well written and provides some interesting findings thus I only have some minor comments.

Please add some values to the results section. Even though you have included the statistical results in the tables I think its a bit easier to follow if you include eg effect sizes or means when you talk about small or moderate changes.

Why have you analysed the teams seperatly instead of including a dummy variable per team to the model?

7. PLOS authors have the option to publish the peer review history of their article (what does this mean?). If published, this will include your full peer review and any attached files.

Reviewer #1: No

Reviewer #2: No

---

## [Decision Letter · Decision Letter 2]

6 Dec 2021

The impact of matches and travel on rugby players’ sleep, wellness and training.

PONE-D-21-08417R2

Dear Dr. Stewart,

We’re pleased to inform you that your manuscript has been judged scientifically suitable for publication and will be formally accepted for publication once it meets all outstanding technical requirements.

Kind regards,

Daniel Boullosa

Academic Editor

PLOS ONE

Additional Editor Comments (optional):

Reviewers' comments:

Reviewer's Responses to Questions

**Comments to the Author**

1. If the authors have adequately addressed your comments raised in a previous round of review and you feel that this manuscript is now acceptable for publication, you may indicate that here to bypass the “Comments to the Author” section, enter your conflict of interest statement in the “Confidential to Editor” section, and submit your "Accept" recommendation.

Reviewer #1: All comments have been addressed

Reviewer #2: All comments have been addressed

2. Is the manuscript technically sound, and do the data support the conclusions?

Reviewer #1: Yes

Reviewer #2: Yes

3. Has the statistical analysis been performed appropriately and rigorously? 

Reviewer #1: Yes

Reviewer #2: Yes

4. Have the authors made all data underlying the findings in their manuscript fully available?

Reviewer #1: Yes

Reviewer #2: Yes

5. Is the manuscript presented in an intelligible fashion and written in standard English?

Reviewer #1: Yes

Reviewer #2: Yes

6. Review Comments to the Author

Reviewer #1: (No Response)

Reviewer #2: (No Response)

7. PLOS authors have the option to publish the peer review history of their article (what does this mean?). If published, this will include your full peer review and any attached files.

Reviewer #1: No

Reviewer #2: No

---

## [Editor Report · Acceptance letter]

31 Jan 2022

PONE-D-21-08417R2 

The impact of matches and travel on rugby players’ sleep, wellness and training. 

Dear Dr. Stewart:

I'm pleased to inform you that your manuscript has been deemed suitable for publication in PLOS ONE. Congratulations! Your manuscript is now with our production department. 

Kind regards, 

on behalf of

Dr. Daniel Boullosa 

Academic Editor

PLOS ONE